

**Establishing relationship between measured and predicted soil water characteristics**
**using SOILWAT model in three agro-ecological zones of Nigeria**
**OrevaOghene Aliku\* and Suarau O. Oshunsanya**
**Department of Agronomy, University of Ibadan, Nigeria**
**\*Corresponding Author: orevaoghenealiku@gmail.com**
**Abstract**
Soil available water (SAW) affects soil nutrients availability and consequently affects crop
performance. However, field determination of SAW for effective irrigated farming is
laborious, time consuming and expensive. Therefore, experiments were initiated at three
agro-ecological zones of Nigeria to compare the measured laboratory and predicted soil
available water using SOILWAT model for sustainable irrigated farming.
One hundred and eighty soil samples were collected from the three agro-ecological zones
(Savannah, Derived savannah and rainforest) of Nigeria and analysed for physical and
chemical properties. Soil texture and salinity were imputed into SOILWAT model (version
6.1.52) to predict soil physical properties for the three agro-ecological zones of Nigeria.
Measured and predicted values of field capacity, permanent wilting point and soil available
water were compared using T-test.
Predicted soil textural classes by SOILWAT model were similar to the measured laboratory
textural classes for savannah, derived savannah and rainforest zones. However, bulk density,
maximum water holding capacity, permanent wilting point and soil available water were
poorly predicted as significant ($p<0.05$) differences existed between measured and predicted
values. Therefore, SOILWAT model could be adopted for predicting soil texture for
savannah, derived savannah and rainforest zones of Nigeria. However, the model needs to be
upgraded in order to accurately predict soil water characteristics of the aforementioned
locations for sustainable irrigation planning.
**Keywords:** Field capacity, permanent wilting point, soil available water, soil texture,
SOILWAT model





## 1.0    Introduction

Water holding capacity is very important for assessing the water demand of vegetation, as
well as for the recharge of the ground water storage. However, irregularities in rainfall
amount and distribution resulting from the advent of climate, and intensive cultivation with
severe erosion degradation have led to a decline in available land for crop production. Soil
water is a basic requirement for plants survival because soil water determines to a very large
extent the availability of plant nutrients to crops. Therefore, change in the soil water within a
given soil profile or across a given landscape play a central role in soil available water, water
conductivity, irrigation scheduling, drainage, evapotranspiration and the transport of salts and
fertilizers.
As a result, several methods have been developed to estimate soil water characteristics of
different types of soils for different agro-ecologies. Though, farmers in rural areas cultivate
various crops by guessing the available moisture content of the soil by means of observation
and feeling methods, one of the major drawbacks with this method is that the estimation of
soil moisture is subjective and not exact (Schneekloth et al., 2007). Saxton and Rawls (2006)
noted that estimation of soil water requirements would require soil water infiltration,
conductivity, storage, and plant-water relationships. Common scientific methods of
estimating soil water requirement involve direct or indirect determination in the laboratory.
These methods use measurements or indicators of water content or a physical property that is
sensitive to changes in water content.
On the other hand, laboratory methods of determining soil available water are costly and time
consuming. Difficulty in describing the mechanical behaviour and water characteristics of
soils has led to the often use of models with different approaches for monitoring soil moisture
conditions (Van Genuchten and Leij, 1992). Guswa et al. (2002) reported that simple models
for soil moisture dynamics, which do not resolve spatial variations in saturation, facilitates
analytical expressions of soil and plant behaviour as functions of climate, soil and vegetation
characteristics. Application of this knowledge is imperative for simulation of soil
hydrological properties within natural landscapes. The Soil Water Characteristics Program
(SOILWAT model) developed by Keith Saxton and Walter Rawls in cooperation with the
Department of Biological Systems Engineering, Washington State University (Oyeogbe et
al., 2012), estimates soil water potential, conductivity and water holding capability based on
soil properties such as texture, organic matter, gravel, salinity, and compaction. The texture
based method reported by Saxton et al. (1986) was largely based on the data set and analyses



of Rawls et al. (1982), who successfully applied the texture based method to a wide variety of
analyses, particularly those of agricultural hydrology and water management using the
SOILWAT model (Saxton and Willey, 2006). Other methods have provided similar results
but with limited versatility (Stolte et al., 1994). Saxton and Rawls (2006) reported that
estimating soil water hydraulic characteristics from readily available physical parameters has
been a long-term goal of soil physicists and engineers. They further reported that many early
trials were sufficiently successful with limited data sets to suggest that there were significant
underlying relationships between soil water characteristics and parameters such as soil
texture (Ahuja et al., 1999; Gijsman et al., 2002).
Recently, validation of the soil water characteristic model by comparing its predicted values
with laboratory determined values have been based on soil texture and organic matter (Saxton
and Rawls, 2006) at a particular soil depth within site(s) (Oyeogbe and Oluwasemire, 2013).
Extrapolation of soil hydrological parameters predicted for a particular environment to
another environment may be misleading due to differences in soil properties (soil
heterogeneity). According to Guswa et al. (2002), proper application of models requires
knowledge of the conditions under which the underlying simplifications are appropriate.
Therefore, this study was carried out to compare laboratory and predicted SOILWAT model
values of soil available water for sustainable irrigated farming in the three agro-ecological
zones of Nigeria.

## 86    2.0    Materials and methods

### 87    2.1    Study site

The study was conducted in three agro-ecological zones of Nigeria; derived savannah – Ogun
State (latitude 05° 41′ N and longitude 06° 03′ E); savannah – Kogi State (latitude 06° 49′ N
and longitude 06° 11′ E) and rainforest – Edo State (latitude 06° 41′ N and longitude 06° 36′
E). According to the international systems of soil classification FAO-UNESCO-ISRIC
(1990), soils from Ogun State developed from Sedimentary rock while soils from Kogi State
and Edo State developed from Basement complex rocks.

**95    Derived Savannah (Ogun state)**

This is predominantly grassy vegetation with a few scattered fire-resistant woody trees and
date palm. It occupies an area of 493.36 ha. The soils are well drained and have a slope ≤ 2%.
The mean rainfall of 1150 mm/year and the temperature range of 20–35°C. Derived savannah



is believed to have developed in areas which have been over-cultivated or subject to
persistent burning especially during the dry season but the derived savannah in the area was
brought about as a result of persistent high water table. The most common grasses in the
project area are *Panicum maximum* and *Imperata cylindrical*. *Pennisetum purpureum* is
found on the poorly drained areas or seasonally swampy areas. Very few scattered poor
stands of cocoa and kola nuts were found in the area. Arable crops like cassava, yam, maize
and leafy vegetables were found in the well-drained part of the project area. The soils are
found over sedimentary rocks in Western Nigeria. Majority of the soils are formed over
leached sandstone, without hard pan and with mottled clay. The parent materials (sandstones)
are fairly well consolidated with mudstone bands, of Eocene age, Cretaceous age or loosely
consolidated sandstones of Tertiary post-Eocene age. They are more or less ferruginous.
These soils are generally termed "Acid sands" because of the sandy parent materials from
which they are formed.
**Savannah**
This agro-ecological zone has a mean rainfall of 1200 – 1400 mm/year. It has a temperature
range of 22 – 33°C. The soils are fairly drained and are formed from crystalline basement
complex rocks. The project area occupies an area of 69.83ha and has a slope ≤ 4.5%. The
type of vegetation is secondary forest.
**Rainforest**
The humid rainforest agro-ecological zone has a mean rainfall of 1200 mm/year with a
temperature of 15 – 34°C. The soils are alluvial kandiudult deposits of River Niger, formed
from underlying basement complex rocks. The sols are poorly drained and have a slope of 2 –
3%. The project area is 305.25 ha. The type of vegetation is secondary forest which consists
of tree crops such as oil palm.

**2.2    Soil sampling**
Four modal soil profile pits (150 – 200 cm deep) were sank at each mapping unit after soil
identification and mapping was done by the rigid grid method. Soil samples were collected
with the aid of soil auger from 0 – 30 cm and 30 – 60 cm (subsurface) of each profile,
respectively. The profiles were described following FAO guidelines (FAO, 2006) at the agro-
ecological zones of Nigeria.
**2.3    Soil analysis**
Composite samples were analysed for physical and chemical properties. Electrical
conductivity was determined with a Conductivity Bridge in a 1:2 soil/water extract (Mclean,



1982). Soil pH was read from an EEL pH meter with glass electrodes inserted into 1:1
soil/water suspension (Mclean, 1982). Organic carbon was determined by the Walkley-Black
dichromate titration method (Nelson and Sommers, 1982). Particle size analysis was by
hydrometer method (Gee and Or, 2002), using sodium hexametaphosphate as dispersing
agent. The functional relationship between soil wetness and matric suction was determined
by means of a tension-table assembly in low suction range (<0.07 bars), and pressure plate
apparatus for the higher tension range (1 to 15 bars) (Hillel, 1971). Bulk density was
measured by the core method in which core samples were oven-dried at 105°C until a
constant weight was achieved. The dry weight of the soil was expressed as the fraction of the
volume of the core as described by Grossman and Reinsch (2002).

**2.4    SOILWAT model description**
The Soil Water Characteristics Program (SOILWAT model) is a predictive system that was
programmed for a graphical computerised model to provide easy application and rapid
solutions in hydrologic analyses (Saxton and Rawls, 2006). The predictive equations used for
the SOILWAT model were generated using an extensive laboratory data set of soil water
characteristics obtained from the USDA/NRCS National Soil Characterisation database (Soil
Survey Staff, 2004). The data included soil water content at 33- and 1500-kPa tensions; bulk
densities; sand (S), silt and clay (C) particle sizes; and organic matter, that were developed
with standard laboratory procedures (USDA-SCS, 1982).

According to Saxton and Rawls (2006), regression equations were then developed for
moisture held at tensions of 1500, 33, 0 to 33 kPa, and air-entry tensions. Air-entry values
were estimated using the exponential form of the Campbell equation (Rawls et al., 1992),
while saturation moisture ($\theta_s$) values were estimated from the reported sample bulk densities
assuming a particle density value of 2.65 g cm$^{-3}$ (Saxton and Rawls, 2006). The new moisture
tension equations were combined with conductivity equations of Rawls et al. (1998) and
additional equations for density, gravel, and salinity effects (Saxton and Rawls, 2006). They
further reported that the resultant equations were then compared with three independent data
sets representative of a wide range of soils to verify their capability for field applications. The
new predictive equations used by the SOILWAT model to estimate soil water content at
selected tensions of 1500, 33, 0 to 33, and $\psi_e$ kPa are summarized in Table 1, while the
symbols for the parameters are defined in Table 2 (Saxton and Rawls, 2006).



The derived equations were incorporated into the graphical computer program to readily
estimate soil hydrological characteristics. The predictive system (SOILWAT graphical
computerised model) is available at http://hydrolab.arsusda.gov/soilwater/Index.htm.

**2.5     Model application**
The values for the independent and dependent variables were obtained and tabulated. The
independent variables were percentage sand, percentage clay, percentage organic matter,
percentage gravel, salinity, and compaction while dependent variables were wilting point,
field capacity, available water, saturated hydraulic conductivity, saturation and bulk density.
The derived independent variables were incorporated into the SOILWAT graphical computer
program to estimate water holding and transmission characteristics (Fig. 1). Texture was
selected from the textural triangle and slider bars were adjusted for organic matter, salinity,
gravel, and compaction. The results were dynamically displayed in text boxes and on a
moisture-tension and moisture-conductivity graph (Fig. 1) as the inputs were varied.
**2.6     Statistical analysis**
Data from observed and predicted methods were subjected to t-test statistic using the GenStat
statistical software (8[th] Edition). Soil moisture content at selected tensions of wilting point,
field capacity, saturation and available water were also subjected to polynomial regression.

**3.0     Results and Discussion**
**3.1     Soil texture and salinity of different depths of the study area**
The soil texture and salinity status at the time of sampling are presented in Table 3, showing
the particle size distribution down the profile. The results from laboratory analysis indicated
an increase in the clay content and a decrease in the sand content down the depth in Savannah
and Derived savannah, while rainforest had a decrease in clay content and an increase in sand
content down the depth. At the depth of 0 to 60 cm, the clay content increased from 6.75 to
14.9% in Savannah, 19.07 to 35.35% in derived savannah, and decreased in rainforest from
26.2 to 17.3%. However, the sand fraction decreased from 92 to 84.2% and 76.6 to 61.3% in
savannah and derived savannah, respectively, while in rainforest there was an increase in
sand content from 64.2 to 78.4%. The surface soils varied from loamy sand to sandy clay
while the subsurface textures had a marginal change from sandy clay loam to sandy clay
among the three agro-ecological zones of Nigeria. Salinity level was lower in surface soils
(0.07 dS m$^{-1}$) than subsurface soils (0.23 dS m$^{-1}$) in savannah, while the reverse was the case




of derived savannah and rainforest. The result of the particle size distribution showed the
dominance of sand sized particles in the three locations. With the exception of rainforest
zone, the higher values of sand compared to silt and clay fractions is typical of soils in
savannah and derived savannah agro-ecological zones of Nigeria (Babalola et al., 2000).
Chris-Emenyonu and Onweremadu (2011) reported that these soils are formed largely from
the coastal plain sands. Contrary to Ogeh and Ukodo (2012) silt content was found to
decrease with increase in depth in all the agro-ecological zones. In the rainforest zone, the
clay content was found to decrease with increase in depth as opposed to savannah and
derived savannah zones. This result is in line with Ogeh and Ukodo (2012) who reported that
the movement of clay through the process of illuviation may be responsible for the high clay
content in the top soils of this region.

**3.2    Soil Available Water**
Table 4 showed the values of soil available water from the laboratory were significantly
higher ($p < 0.05$) than those predicted by the model in all the locations, indicating that
SOILWAT model did not accurately predict soil available water for savannah, derived
savannah and rainforest, respectively. In savannah, laboratory soil available water values
increased with depth from 3.77 to 9.41 cm, while the predicted value was 0.07 cm at the
corresponding depths. In derived savannah, both laboratory and predicted soil available water
(SAW) values increased with increase in depth. Laboratory SAW values increased from 4.71
to 9.38 cm and the predicted SAW values increased from 0.07 to 0.08 cm. However, in
rainforest, there was increase in the laboratory SAW values from 3.21 to 8.15 cm, while the
predicted SAW values decreased from 0.08 to 0.06 cm with depth. The best regression for
available water was obtained for soils in derived savannah ($R^2 = 0.44$) indicating that SAW
could be predicted using SOILWAT model (Fig. 2). However, savannah ($R^2 = 0.25$) and
rainforest ($R^2 = 0.13$) had poor regression between laboratory and predicted SAW, suggesting
that the SOILWAT model had poor SAW prediction for the aforementioned locations. These
results may be due to the exclusion of organic matter data in the model adjustments, which
could influence soil water. Saxton and Rawls (2006) stated that organic matter content of the
soil play a major role in soil water retention.
**3.3    Bulk Density**
The values for measured (laboratory) and predicted bulk density are summarised in Table 5.
Values obtained from the laboratory (1.31 and 1.41 g cm$^{-3}$) were significantly lower ($p < 0.05$)



than the predicted values (1.66 and 1.55 g cm$^{-3}$) for savannah at $0-30$ cm and $30-60$ cm
depths. However, derived savannah and rainforest bulk density values from the laboratory
were lower at $0-30$ cm depth and higher at $30-60$ cm depth than the predicted values. It
was noted that bulk density values were higher in soils from $30-60$ cm depth than $0-30$ cm
depth for all locations. This could be ascribed to increase in soil compaction down the soil
profile. Soil compaction has been reported to be associated with increase in bulk density
which is one of the soil physical properties that may affect crop growth and yield (Lipiec et
al., 1991; Lowery and Schuller, 1994; Mamman and Ohu, 1997). However, the predicted
bulk density values at $30-60$ cm depth was lower than $0-30$ cm depth in savannah and
derived savannah. This could be due to the absence of silt adjustments in the SOILWAT
programmed textural triangle. Saxton and Rawls (2006) reported that the density values at the
texture extremes (sands and clays) may be most likely to require adjustments. There was no
significant difference between the observed and predicted bulk density values in rainforest
zone.

### 3.4    Field Capacity

The measured field capacity values were lower than the predicted values in all the three
locations (Table 6). Both measured and predicted field capacity values in savannah zone
increased from 13.5 to 15.0%, and 13.9 to 18.3% respectively for $0-60$ cm depth. However,
derived savannah soils showed a decrease in the measured field capacity values from 21.34 to
18.78%, while the predicted values increased from 22.01 to 29.59% with depth. Both the
measured and predicted values were not significant at $0-30$ cm but decreased from 18.10 to
14.86% (measured) and 28.40 to 20.75% (predicted) in the rainforest zone. Figure 3 showed
that the regression for field capacity with both $0-30$ cm and $30-60$ cm depth data in all
locations were poor ($R^2 = 0.20$). These results do not agree with Saxton and Rawls (2006)
who reported higher $R^2$ value of 0.63 due to the inclusion of appropriate local adjustments for
organic matter, density and gravel in addition to salinity. They further reported that field
capacity values will be most affected by organic matter adjustments, which has been reported
to enhance soil water retention because of its hydrophilic nature and its positive influence on
soil structure (Huntington, 2007).

### 3.5    Hydraulic Conductivity

The measured and predicted values for soil hydraulic conductivity under the three locations
are summarised in Table 7. Measured values of 18.8 and 18.1 cm s$^{-1}$ (savannah); 10.1 and 9.7



cm s$^{-1}$ (derived savannah); and 8.7 and 8.6 cm s$^{-1}$ (rainforest) were significantly (p<0.05)
higher than the predicted values of 4.8 and 1.3 cm s$^{-1}$ (savannah); 0.6 and 0.2 cm s$^{-1}$ (derived
savannah); 0.4 and 1.0 cm s$^{-1}$ (rainforest) at 0 – 30 cm and 30 – 60 cm soil depths,
respectively. Both the measured and predicted hydraulic conductivity values for savannah
and derived savannah were higher in 0 – 30 cm depth than 30 – 60 cm depth. This could be
attributed to the increase in soil compaction down the profile. However, predicted saturated
hydraulic conductivity for 0 – 30 cm depth was higher than 30 – 60 cm depth in rainforest. It
also revealed that both measured and predicted hydraulic conductivity values decreased with
soil depth in all locations except the predicted values which increased in rainforest zone. The
significant difference between the predicted and measured SHC values may be due to the
unavailability of soil density data for the simulation process. Carman (2002) reported that soil
density affects the physical, mechanical and hydraulic properties of soils. Saxton and Rawls
(2006) stated that soil density strongly affects soil structure and large pore distribution,
consequently affecting saturated hydraulic conductivity. They further reported that a change
in density factor will largely affect saturated hydraulic conductivity.
**3.6    Moisture Content (MC)**
Measured and predicted MC values are depicted in Table 8. The results showed that the
measured MC values (18.79 and 18.87%) were higher than the predicted (9.56 and 11.41%)
values in savannah soils. However, measured MC values of soils from derived savannah and
rainforest were found to be lower than the predicted values. Measured MC values were 4.80
and 9.52% (derived savannah); and 3.40 and 9.36% (rainforest), while the predicted values
were 14.71 and 21.32% (derived savannah); and 20.90 and 15.04% (rainforest) at 0 – 30 cm
and 30 – 60 cm soil depths, respectively. Both measured and predicted MC values were
significantly (P<0.05) higher in all locations at 30 – 60 cm depth, with the exception of
rainforest zone. Several estimating methods developed in recent years have shown that
generalized predictions can be made with usable, but variable accuracy (Rawls *et al.*, 1982;
Saxton et al., 1986; Stolte et al., 1994). Meissner (2004) reported a similar result that the
inclusion of bulk density as an input to their model work improved the accuracy of soil water
content estimation.
**3.7    Maximum water holding capacity (MWHC)**
Table 9 showed that the measured MWHC values were significantly (p<0.05) lower than the
predicted MWHC values in all locations. Soils from savannah zone had the measured
MWHC values of 18.85 and 18.56% and predicted MWHC values of 37.21 and 41.69%.





However, derived savannah had measured MWHC values of 24.45 and 20.92% and predicted
MWHC values of 44.33 and 48.44% while rainforest zone had observed MWHC values of
20.17 and 16.88% and predicted MWHC values of 46.97 and 42.95% at 0 – 30 cm and 30 –
60 cm soil depths, respectively. The graphical results of regression for MWHC are shown in
Figure 4 for all locations. The best regression graph was obtained for soils in savannah ($R^2$ =
0.45), followed by derived savannah ($R^2$ = 0.13) and least by rainforest ($R^2$ = 0.05). This may
be due to the fact that MWHC values may be based on factors which have no relationship
with the correlation variables of texture. A similar result was also reported by Saxton and
Rawls (2006) who reported that preliminary regression results for MWHC with two horizon
data were poor ($R^2$ = 0.25). Rawls (1983) and Grossman et al. (2001) explained that the poor
regression result of the tested values may be due to the influence of factors such as tillage,
root and worm activities, which are not part of the input parameters of the model.
**3.8    Wilting Point (WP)**
The laboratory measured and predicted WP values for the three locations are summarized in
Table 10. The measured WP values were found to be significantly lower than the predicted
values at $p<0.05$ in all locations. Soils from 0 – 30 cm and 30 – 60 cm depth in savannah had
observed WP values of 1.07 and 2.80%, while the predicted values were 7.25 and 11.25%,
respectively, while observed WP values for derived savannah (2.81 and 5.45%) and rainforest
(4.80 and 3.44%) were also lower than their respective predicted values at 0 – 30 cm and 30 –
60 cm depths, respectively. Both the measured and predicted WP values were higher at the 30
– 60 cm soil depth in soils from savannah and derived savannah, while soils from rainforest
had lower values at 30 – 60 cm soil depth. Figure 5 showed that the best wilting point
regression was in savannah ($R^2$ = 0.84), followed by derived savannah ($R^2$ = 0.66) and least
by rainforest ($R^2$ = 0.09). The result obtained in savannah is in line with Saxton and Rawls
(2006) who reported $R^2$ value of 0.86. They obtained the best regression with wilting point by
using regression deviations as a guide in addition to slight adjustments of the clay content.
**4.0    Conclusions**
The SOILWAT model provides a quick visual display of the predicted textural classes that
are similar to laboratory determined textural classes for savannah, derived savannah and
rainforest zones of Nigeria. Also, the regression equations used to validate the integrity of the
model parameters were strong for wilting point in the savannah and derived savannah agro-
ecological zones. Results further showed that soil texture alone is not sufficient to predict soil
water characteristics. However, additional variables such as organic matter, bulk density,



gravel and salinity are needed for accurate prediction of soil water parameters. In addition,
measured and predicted variables (field capacity, wilting point and soil available water ) were
significantly ($p < 0.05$) different, suggesting that SOILWAT model needs some improvements
for better prediction of soil moisture characteristics for irrigation planning and scheduling.

**Code and/or data availability**

SOILWAT model is a graphical computerised program developed with a predictive system
that enhances the opportunity to integrate information on soil water characteristics into
hydrologic analysis and water management decisions. It is available at
http://hydrolab.arsusda.gov/soilwater/Index.htm, while the new predictive equations used to
estimate soil water content in the model can be obtained from Saxton and Rawls (2006).

**Author contribution**

S. O. Oshunsanya designed the experiment while OrevaOghene Aliku carried it out and
performed the simulation. OrevaOghene Aliku prepared the manuscript with contributions
from S. O. Oshunsanya who also edited the manuscript.

**Acknowledgement**

The authors acknowledge the contributions of the laboratory technologist in the Department
of Agronomy, University of Ibadan for their assistance in the various soil physical and
hydrological analysis.

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

Riverside, CA.





Table 1: Summary of equations for soil water characteristics estimates.†

| Variable | Equation | $R^2/S_e$ | Eq. |
|---|---|---|---|
| **Moisture Regressions** | | | |
| $\theta_{1500}$ | $\theta_{1500} = \theta_{1500t} + (0.14 \times \theta_{1500t} - 0.02)$ | 0.86/0.02 | 1 |
| | $\theta_{1500t} = -0.024S + 0.487C + 0.006OM + 0.005(S \times OM) - 0.013(C \times OM) + 0.068(S \times C) + 0.031$ | | |
| $\theta_{33}$ | $\theta_{33} = \theta_{33t} + [1.283(\theta_{33t})^2 - 0.374(\theta_{33t}) - 0.015]$ | 0.63/0.05 | 2 |
| | $\theta_{33t} = -0.251S + 0.195C + 0.011OM + 0.006(S \times OM) - 0.027(C \times OM) + 0.452(S \times C) + 0.299$ | | |
| $\theta_{S-33}$ | $\theta_{S-33} = \theta_{(S-33)t} + (0.636\theta_{(S-33)t} - 0.107)$ | 0.36/0.06 | 3 |
| | $\theta_{(S-33)t} = 0.278S + 0.034C + 0.022OM - 0.018(S \times OM) - 0.027(C \times OM) - 0.584(S \times C) + 0.078$ | | |
| $\psi_e$ | $\psi_e = \psi_{et} + (0.02\psi_{et}^2 - 0.113\psi_{et} - 0.70)$ | 0.78/2.9 | 4 |
| | $\psi_{et} = -21.67S - 27.93C - 81.97\theta_{S-33} + 71.12(S \times \theta_{S-33}) + 8.29(C \times \theta_{S-33}) + 14.05(S \times C) + 27.16$ | | |
| $\theta_S$ | $\theta_S = \theta_{33} + \theta_{(S-33)} - 0.097S + 0.043$ | 0.29/0.04 | 5 |
| $\rho_N$ | $\rho_N = (1 - \theta_S)2.65$ | | 6 |
| **Density Effects** | | | |
| $\rho_{DF}$ | $\rho_{DF} = \rho_N \times DF$ | | 7 |
| $\theta_{S-DF}$ | $\theta_{S-DF} = 1 - (\rho_{DF}/2.65)$ | | 8 |
| $\theta_{33-DF}$ | $\theta_{33-DF} = \theta_{33} - 0.2(\theta_S - \theta_{S-DF})$ | | 9 |
| $\theta_{(S-33)DF}$ | $\theta_{(S-33)DF} = \theta_{S-DF} - \theta_{33-DF}$ | | 10 |
| **Moisture-Tension** | | | |
| $\psi_{(1500-33)}$ | $\psi_\theta = A(\theta)^{-B}$ | | 11 |
| $\psi_{(33-\psi e)}$ | $\psi_\theta = 33.0 - [(\theta - \theta_{33})(33.0 - \psi_e)/(\theta_S - \theta_{33})]$ | | 12 |
| $\theta_{(\psi e-0)}$ | $\theta = \theta_s$ | | 13 |
| A | $A = \exp(\text{In}33 + B\text{In}\theta_{33})$ | | 14 |
| B | $B = [\text{In}(1500) - \text{In}(33)]/[\text{In}(\theta_{33}) - \text{In}(\theta_{1500})]$ | | 15 |
| | Moisture-Conductivity | | |
| $K_S$ | $K_S = 1930(\theta_s - \theta_{33})^{(3-\lambda)}$ | | 16 |
| $K_\theta$ | $K_\theta = K_s(\theta/\theta_s)^{[3+(2/\lambda)]}$ | | 17 |
| $\Lambda$ | $\lambda = 1/B$ | | 18 |
| | Gravel Effects | | |
| $R_v$ | $R_v = (\alpha R_w)/[1 - R_w(1 - \alpha)]$ | | 19 |
| $\rho_B$ | $\rho_B = \rho_N(1 - R_v) + (R_v \times 2.65)$ | | 20 |
| $PAW_B$ | $PAW_B = PAW(1 - R_v)$ | | 21 |
| $K_b/K_s$ | $K_b/K_s = \dfrac{1 - R_w}{[1 - R_w(1 - 3\alpha/2)]}$ | | 22 |
| **Salinity Effects** | | | |
| $\Psi_O$ | $\Psi_O = 36EC$ | | 23 |
| $\Psi_{O\theta}$ | $\Psi_{O\theta} = \dfrac{\theta_S(36EC)}{\theta}$ | | 24 |

† All symbols are defined in Table 2. The coefficient of determination ($R^2$) and standard error
of estimate ($S_e$) define the data representation and expected predictive accuracy.
Source: Saxton and Rawls (2006)







Table 2: Definitions for soil moisture characteristics equation symbols

| Symbol | Definition |
| --- | --- |
| A, B | Coefficients of moisture-tension, Eq. [11] |
| C | Clay, %w |
| DF | Density adjustment Factor (0.9–1.3) |
| EC | Electrical conductance of a saturated soil extract, dS m$^{-1}$ (dS/m = mili-mho cm$^{-1}$) |
| FC | Field Capacity moisture (33 kPa), %v |
| OM | Organic Matter, %v |
| PAW | Plant Available moisture (33–1500 kPa, matric soil), %v |
| $PAW_B$ | Plant Available moisture (33–1500 kPa, bulk soil), %v |
| S | Sand, %w |
| SAT | Saturation moisture (0 kPa), %v |
| WP | Wilting point moisture (1500 kPa), %v |
| $\theta_\psi$ | Moisture at tension $\psi$, %v |
| $\theta_{1500t}$ | 1500 kPa moisture, first solution, %v |
| $\theta_{1500}$ | 1500 kPa moisture, normal density, %v |
| $\theta_{33t}$ | 33 kPa moisture, first solution, %v |
| $\theta_{33}$ | 33 kPa moisture, normal density, %v |
| $\theta_{33-DF}$ | 33 kPa moisture, adjusted density, %v |
| $\theta_{(S-33)t}$ | SAT-33 kPa moisture, first solution, %v |
| $\theta_{(S-33)}$ | SAT-33 kPa moisture, normal density, %v |
| $\theta_{(S-33)DF}$ | SAT-33 kPa moisture, adjusted density, %v |
| $\theta_S$ | Saturated moisture (0 kPa), normal density, %v |
| $\theta_{S-DF}$ | Saturated moisture (0 kPa), adjusted density, %v |
| $\psi_\theta$ | Tension at moisture $\theta$, kPa |
| $\psi_{et}$ | Tension at air entry, first solution, kPa |
| $\psi_e$ | Tension at air entry (bubbling pressure), kPa |
| $K_S$ | Saturated hydraulic conductivity (matric soil), mm h$^{-1}$ |
| $K_b$ | Saturated hydraulic conductivity (bulk soil), mm h$^{-1}$ |
| $K_\theta$ | Unsaturated conductivity at moisture $\theta$, mm h$^{-1}$ |
| $\rho_N$ | Normal density, g cm$^{-3}$ |
| $\rho_B$ | Bulk soil density (matric plus gravel), g cm$^{-3}$ |
| $\rho_{DF}$ | Adjusted density, g cm$^{-3}$ |
| $\lambda$ | Slope of logarithmic tension-moisture curve |
| $\alpha$ | Matric soil density/gravel density (2.65) = $\rho/2.65$ |
| $R_v$ | Volume fraction of gravel (decimal), g cm$^{-3}$ |
| $R_w$ | Weight fraction of gravel (decimal), g g$^{-1}$ |
| $\Psi_O$ | Osmotic potential at $\theta = \theta_S$, kPa |
| $\Psi_{O\theta}$ | Osmotic potential at $\theta < \theta_S$, kPa |

%w = decimal percent by weight basis, %v = decimal percent by volume basis.
Source: Saxton and Rawls (2006)








Table 3: Observed and predicted textural classes for values of sand, silt and clay

| Location | Soil depth | Salinity | Sand | Silt | Clay | Textural class | |
|---|---|---|---|---|---|---|---|
| | (cm) | (dS/m) | Mean weight (%) | | | Laboratory | SOILWAT |
| Savannah | 0 – 30 | 0.07 | 92 | 2.25 | 6.75 | S | S – LS |
| | 30 – 60 | 0.23 | 84.2 | 1.90 | 14.9 | LS – SL | LS – SL |
| Derived savannah | 0 – 30 | 1.02 | 76.57 | 5.79 | 19.07 | LS – SC | LS – C |
| | 30 – 60 | 0.05 | 61.25 | 4.45 | 35.35 | SCL – SC | SCL – SC |
| Rainforest | 0 – 30 | 7.74 | 64.17 | 9.75 | 26.17 | LS – SC | LS – C |
| | 30 – 60 | 5.94 | 78.37 | 4.37 | 17.26 | S – SCL | S – SC |

Note: S: Sand; LS: Loamy sand; SL: Sandy loam; SC: Sandy clay; SCL: Sandy clay loam; C: Clay






Table 4: Comparison of soil available water values from laboratory and SOILWAT model for the three agro-ecological zones of Nigeria

| Location | Depth (cm) | Laboratory | SOILWAT | S.E.± | CV% |
|---|---|---|---|---|---|
| | | Soil available water (%) | | | |
| Savannah | 0 – 30 | 3.77 | 0.07 | 1.43** | 74.30 |
| | 30 – 60 | 9.41 | 0.07 | 3.09** | 65.20 |
| Derived savannah | 0 – 30 | 4.71 | 0.07 | 1.41** | 59.00 |
| | 30 – 60 | 9.38 | 0.08 | 2.68** | 56.60 |
| Rainforest | 0 – 30 | 3.21 | 0.08 | 0.72** | 43.90 |
| | 30 – 60 | 8.15 | 0.06 | 1.89** | 46.0 |

** Significant at p=0.01












Table 5: Comparing bulk density values from laboratory and SOILWAT model for the three agro-ecological zones of Nigeria

| Location | Depth (cm) | Laboratory | SOILWAT | S.E.± | CV% |
|---|---|---|---|---|---|
| | | Bulk density (g cm$^{-3}$) | | | |
| Savannah | 0 – 30 | 1.31 | 1.66 | 0.09** | 6.40 |
| | 30 – 60 | 1.41 | 1.55 | 0.13* | 8.50 |
| Derived savannah | 0 – 30 | 1.28 | 1.48 | 0.16** | 11.30 |
| | 30 – 60 | 1.52 | 1.37 | 0.15** | 10.40 |
| Rainforest | 0 – 30 | 1.34 | 1.41 | 0.14ns | 10.10 |
| | 30 – 60 | 1.57 | 1.51 | 0.17ns | 10.70 |

** Significant at p=0.01; * Significant at p=0.05; ns: not significant at p=0.05


Table 6: Comparing laboratory determined field capacity and SOILWAT predicted field capacity values from agro-ecological zones of Nigeria

| Location | Depth (cm) | Laboratory | SOILWAT | S.E.± | CV% |
|---|---|---|---|---|---|
| | | Field capacity (%) | | | |
| Savannah | 0 – 30 | 13.57 | 13.99 | 2.26ns | 16.40 |
| | 30 – 60 | 15.07 | 18.39 | 2.15** | 12.80 |
| Derived savannah | 0 – 30 | 21.34 | 22.01 | 3.78ns | 17.50 |
| | 30 – 60 | 18.78 | 29.59 | 4.20** | 17.40 |
| Rainforest | 0 – 30 | 18.10 | 28.40 | 6.43** | 27.60 |
| | 30 – 60 | 14.86 | 20.75 | 4.54** | 25.50 |

** Significant at p=0.01; ns: not significant at p=0.05





Table 7: Comparison of laboratory determined saturated hydraulic conductivity values and SOILWAT model for the three agro-ecological zones of Nigeria

| Location | Depth (cm) | Laboratory | SOILWAT | S.E.± | CV% |
|----------|------------|------------|---------|-------|-----|
| | | $Ks$ (cm s$^{-1}$) | | | |
| Savannah | 0 – 30 | 18.80 | 4.80 | 5.80** | 49.20 |
| | 30 – 60 | 18.10 | 1.30 | 5.65** | 58.50 |
| Derived savannah | 0 – 30 | 10.10 | 0.60 | 6.52** | 121.30 |
| | 30 – 60 | 9.70 | 0.20 | 6.41** | 129.50 |
| Rainforest | 0 – 30 | 8.70 | 0.44 | 3.75** | 82.20 |
| | 30 – 60 | 8.64 | 1.09 | 3.71** | 76.3 |

** Significant at p=0.01



Table 8: A comparison of soil moisture content values of laboratory and SOILWAT model for the three ago-ecological zones of Nigeria

| Location | Depth (cm) | Laboratory | SOILWAT | S.E.± | CV% |
|----------|------------|------------|---------|-------|-----|
| | | MC (%) | | | |
| Savannah | 0 – 30 | 18.79 | 9.56 | 2.26** | 16.00 |
| | 30 – 60 | 18.87 | 11.41 | 2.22** | 14.70 |
| Derived savannah | 0 – 30 | 4.80 | 14.71 | 2.88** | 29.60 |
| | 30 – 60 | 9.52 | 21.32 | 4.14** | 26.90 |
| Rainforest | 0 – 30 | 3.40 | 20.90 | 5.23** | 43.10 |
| | 30 – 60 | 9.36 | 15.04 | 4.42** | 36.30 |

** Significant at p=0.01





Table 9: A comparison of laboratory determined and SOILWAT predicted maximum water holding capacity for the three agro-ecological zones of Nigeria

| Location | Depth (cm) | Laboratory | SOILWAT | S.E.± | CV% |
|---|---|---|---|---|---|
| | | MWHC (%) | | | |
| Savannah | 0 – 30 | 18.85 | 37.21 | 3.84** | 13.70 |
| | 30 – 60 | 18.56 | 41.69 | 2.57** | 8.50 |
| Derived savannah | 0 – 30 | 24.45 | 44.33 | 3.22** | 9.30 |
| | 30 – 60 | 20.92 | 48.44 | 3.15** | 9.10 |
| Rainforest | 0 – 30 | 20.17 | 46.97 | 4.15** | 12.30 |
| | 30 – 60 | 16.88 | 42.95 | 3.88** | 13.00 |

** Significant at $p=0.01$



Table 10: A comparison of laboratory determined and SOILWAT predicted values for wilting point for the three agro-ecological zones of Nigeria

| Location | Depth (cm) | Laboratory | SOILWAT | S.E.± | CV% |
|---|---|---|---|---|---|
| | | WP (%) | | | |
| Savannah | 0 – 30 | 1.07 | 7.25 | 1.19** | 28.60 |
| | 30 – 60 | 2.80 | 11.25 | 1.27** | 18.10 |
| Derived savannah | 0 – 30 | 2.81 | 14.70 | 2.53** | 28.90 |
| | 30 – 60 | 5.45 | 21.31 | 3.01** | 22.50 |
| Rainforest | 0 – 30 | 4.80 | 20.70 | 5.12** | 40.10 |
| | 30 – 60 | 3.44 | 14.67 | 4.23** | 46.70 |

** Significant at $p=0.01$








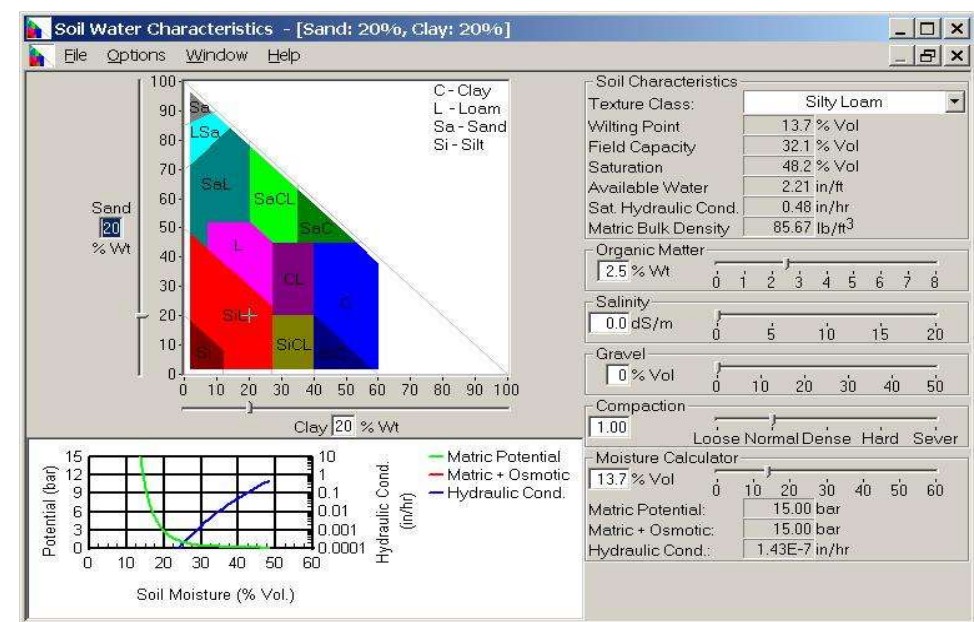

**Fig. 1: Graphical input/output screen of the soil water characteristics model**










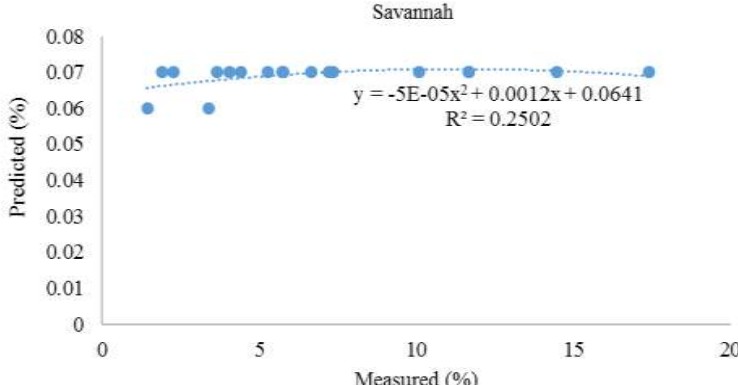


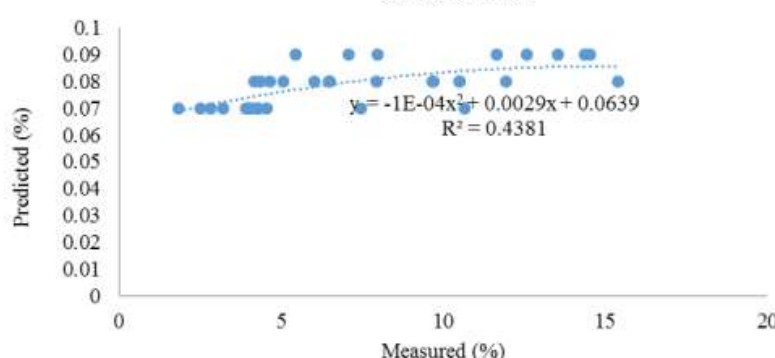


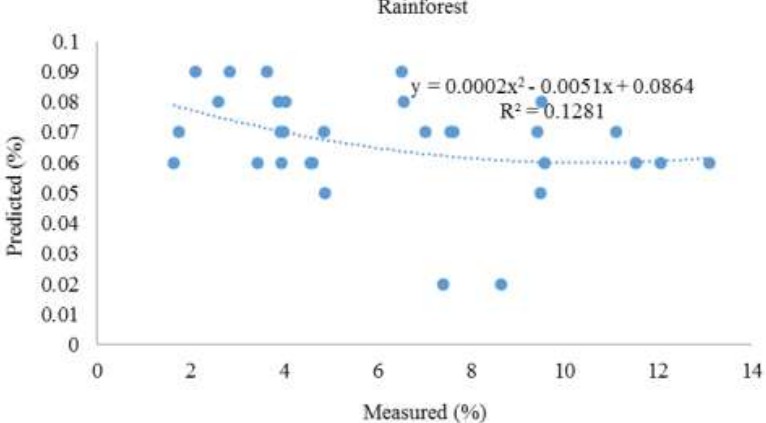


**Fig. 2: Relationship between measured and predicted soil available water expressed by**
**polynomial regression**



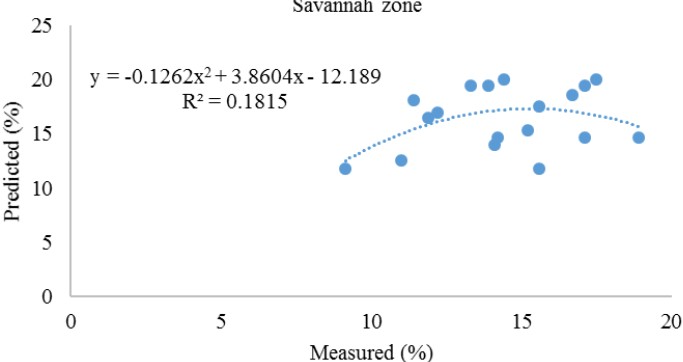


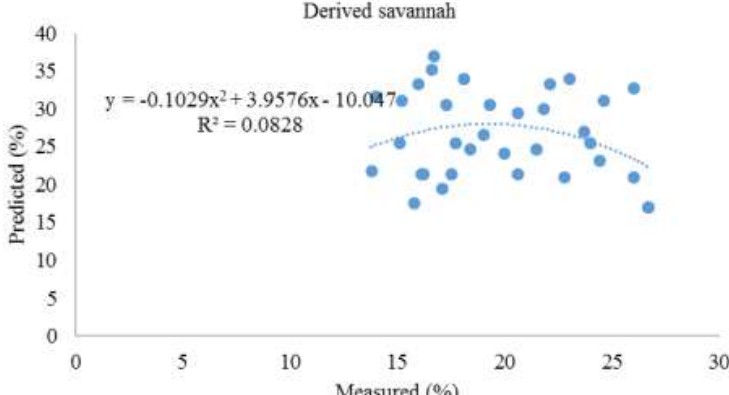


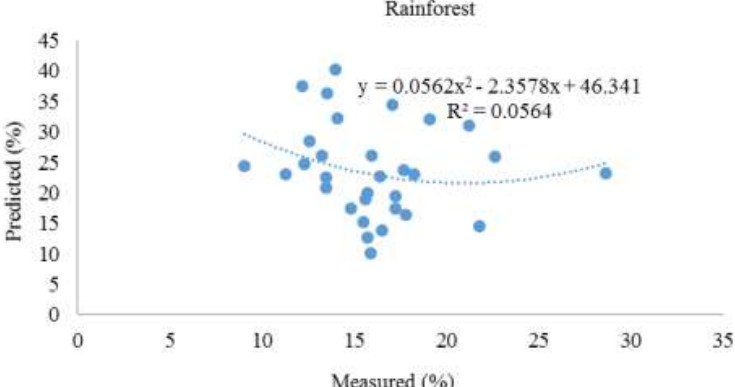


**Fig. 3: Relationship between measured and predicted field capacity expressed by**
**polynomial regression**





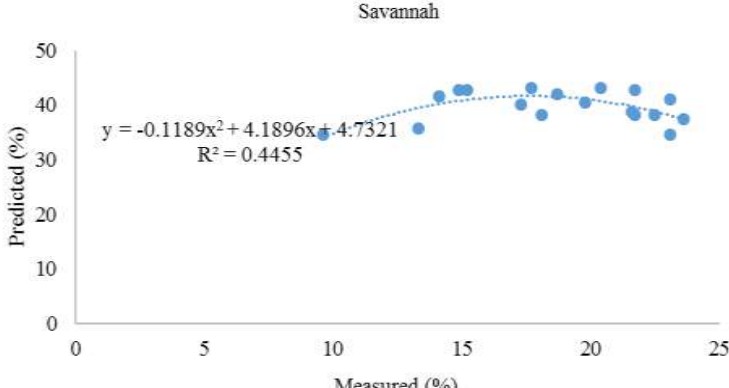


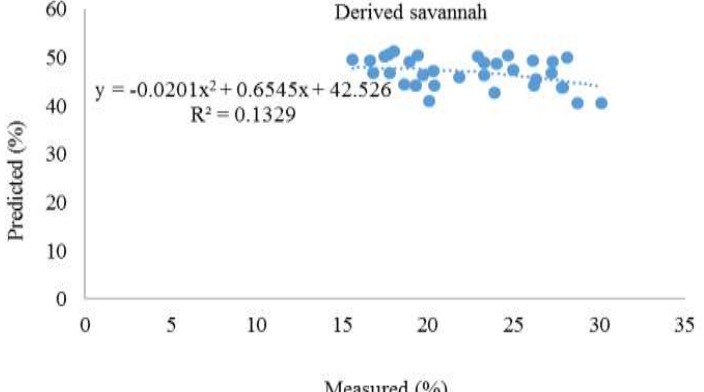



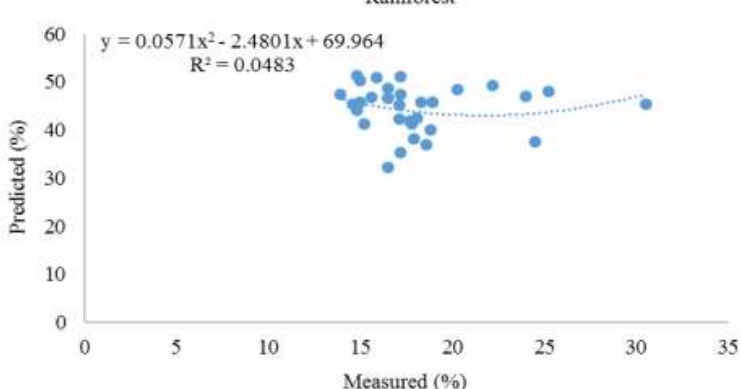


**Fig. 4: Relationship between measured and predicted maximum water holding capacity**
**expressed by polynomial regression**




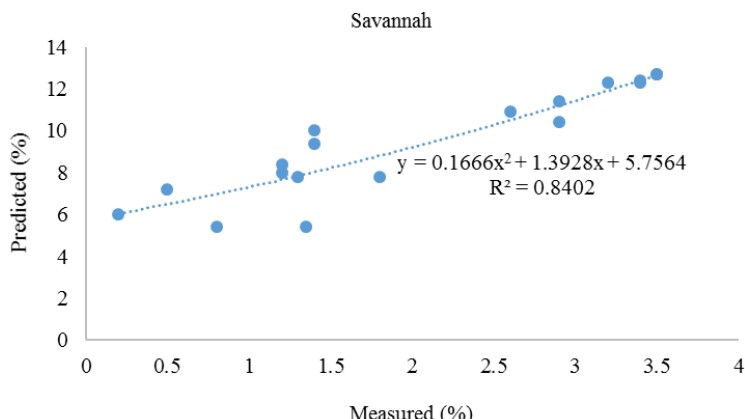


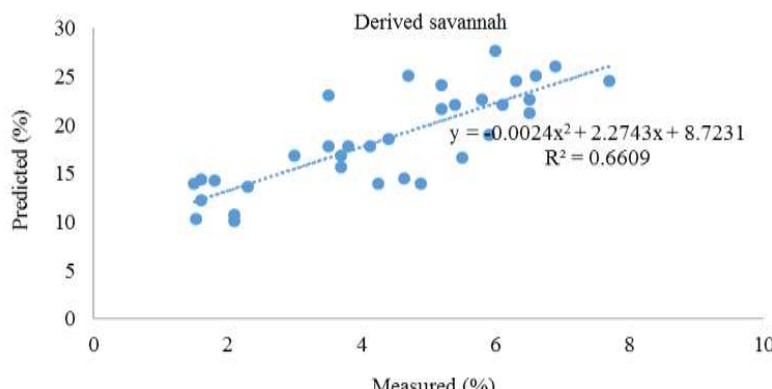


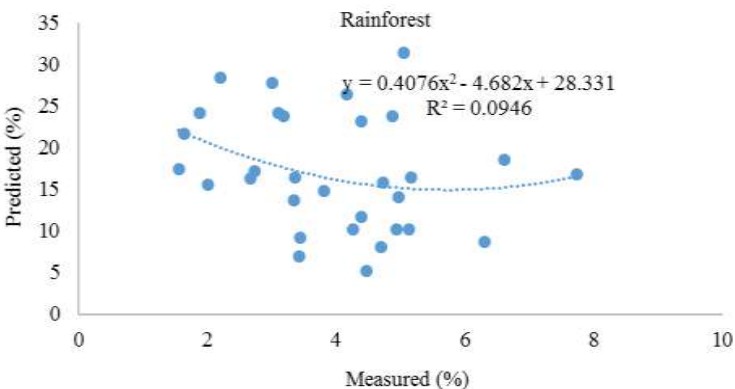


**Fig. 5: Relationship between measured and predicted wilting point expressed by**
**polynomial regression**