# Peer review of "Establishing relationship between measured and predicted soil water characteristics using SOILWAT model in three agro-ecological zones of Nigeria"

_Geoscientific Model Development, 2016_

## Short Comment (SC1) · 22 Aug 2016

Dear authors,

In my role as Executive editor of GMD, I would like to bring to your attention our Editorial version 1.1:

http://www.geosci-model-dev.net/8/3487/2015/gmd-8-3487-2015.html

This highlights some requirements of papers published in GMD, which is also available on the GMD website in the 'Manuscript Types' section:

[Figure]

http://www.geoscientific-model-development.net/submission/manuscript_types.html

In particular, please note that for your paper, the following requirement has not been met in the Discussions paper:

- "The main paper must give the model name and version number (or other unique identifier) in the title."

For a model evaluation it is important to know, which model version exactly was evaluated. Therefore, please add a version number for the SOILWAT model in the title upon your revised submission to GMD.

Yours,

Astrid Kerkweg

――――――――――――――――――

---

## Author Comment (AC1) · 24 Aug 2016

Dear Dr. Astrid Kerkweg,

Thank you for your critical observation sir. The model version number will be added to the manuscript title upon submission of the revised version to GMD.

Thank you for this opportunity of improving the quality of this article. We (the authors) are very grateful.

Warm Regards,

[Figure]

OrevaOghene Aliku

---

## Referee Comment (RC1) · Anonymous Referee #1 · 2 Sep 2016

General comments:

This study compared the measured and predicted soil physical properties (soil available water, bulk density, field capacity, hydraulic conductivity, moisture content, maximum water holding capacity, and wilting point) at three agro-ecological zones of Nigeria. The model examined is the version 6.1.52 of the Soil Water Characteristics Program (SOILWAT model). The motivation behind this study was that soil available water has important impacts on soil nutrients availability, and the use of numerical models to estimate soil physical properties is necessary to avoid time consuming and labor

intensive soil measurements.

However, all soil physical properties predicted by SOILWAT model were significantly different from the measured values at all of the examined agro- ecological zones, which suggests that current version of the SOILWAT model cannot be applied to represent soil water characteristics. Extensive improvements are needed for the SOILWAT model before it can be used for irrigation planning. I suggest rejecting this manuscript at this time.

Specific comments:

Page 2, ln 37-39: Is there any reference that supports this statement? If so, please cite here.

Page 3, ln 96: What type of grassy vegetation is predominant here? C3 or C4 photo-synthesis pathway?

Page 4, ln 112: Please indicate the location of this agro-ecological zone as the format used in "Derived Savannah".

Page 4, ln 117: Please indicate the location of this agro-ecological zone as the format used in "Derived Savannah".

Page 6, ln 181: Which type of the T-test is used in the analysis? One-tailed or two-tailed?

Page 7, ln 221-223: Please rephrase the sentence. The R-squared value (0.44) could be acceptable here, but it doesn't mean that SOILWAT model can be used to predict soil available water.

Page 7, ln 225-228: If model performance relies highly on soil organic matter, why didn't the authors include this information in the SOILWAT model?

Page 8, ln 239-245: Why didn't include silt adjustments in SOILWAT model to improve its performance for bulk density?

Page 8, ln 255-260: Why didn't include appropriate local adjustments for soil organic matter to improve the model results?

Page 9, ln 272-278: I agree with the authors that soil density can largely affect soil hydraulic conductivity simulations, but can it cause an order of magnitude difference between the measured and predicted soil hydraulic conductivity? Was there anything wrong in the SOILWAT model configurations?

Page 10, ln 329-333: The authors claimed that additional variables can help improve the simulation results from SOILWAT model; however, no results were shown to support this statement. More efforts are needed before using the SOILWAT model to predict soil moisture characteristics for irrigation planning and scheduling.

Page 15, Table 1: Please correct the format for "moisture-conductivity" and "gravel effects" (center, bold, and underline).

Page 21, Fig. 1: Please provide a better resolution figure to replace this one. It's hard to tell the characters in each shaded box.
* * *

---

## Referee Comment (RC2) · Anonymous Referee #2 · 10 Sep 2016

General comments:

Soil water is a highly spatially and temporally variable. Determining soil water is quite difficult. Real, direct measurement on soil is valuable. This study use SOILWAT model compared to laboratory measurement of soil physical properties (soil available water, bulk density, field capacity, saturated hydraulic conductivity, soil moisture content, maximum water holding capacity, and wilting point) at three agro-ecological sites of Nigeria.

However, the results showed that SOILWAT cannot act as a good model to present the real status in the case. The authors should pass their own special opinion to modify it.

[Figure]

Otherwise, this work remained uncompleted. For example, in page 7, line 227-228, as Saxton and Rawls (2006) indicated the organic matter content is important, and then the author should make a contribution to resolve it. Investigate the impact of vegetation on soil water is also needed.

Besides, the purpose of this study also wants to manage irrigation planning and scheduling, but only the comparison between model and laboratory measurement has been shown. Not have detailed discussion in the management. The study should take responsibility for the research community, how the works can make a contribution to the related topic should be included.

Some questions and suggestions:

The review in performance of SOILWAT is lacked. How do authors think SOILWAT is a good tool to do such research in this region? Authors should use the physics formula to explain the uncertainties of predicted parameters. The explanations are too weak.

Page 4, line 87: Why do authors choose these sites?

Page 4, line 112: Lack of information as what in 'Derived Savannah'

Page 4, line 117: Lack of information as what in 'Derived Savannah'

It may need a real map to show the location of these three sites.

Page 7, line 222-223: The R-squared value (0.44) could not indicate that SOILWAT model can be used to predict soil available water.

Page 9, line 279: The results showed SOILWAT overestimated at Derived savannah and rainforest but under estimated in Savannah in soil moisture content, how do authors comment it?

Page 21, Figure. 1: Please provide the description on it and also a better resolution one to replace it.

---

## Author Comment (AC2) · 11 Oct 2016

**Revision Note**

**Journal Name:** Geoscientific Model Development

**Manuscript Number:** gmd-2016-165

**Manuscript Title:** Establishing relationship between measured and predicted soil water characteristics using SOILWAT model in three agro-ecological zones of Nigeria

Dear Editor,

Thank you very much for providing me with the opportunity to revise the manuscript. I have revised and improved all sections following your suggestions.

NB: All revised areas have been highlighted in yellow colour in the annotated manuscript (manuscript tagged "Soil Water Characteristics Model Revised").

**ANONYMOUS REFEREE #1**

**General comments**

This study compared the measured and predicted soil physical properties (soil available water, bulk density, field capacity, hydraulic conductivity, moisture content, maximum water holding capacity, and wilting point) at three agro-ecological zones of Nigeria. The model examined is the version 6.1.52 of the Soil Water Characteristics Program (SOILWAT model). The motivation behind this study was that soil available water has important impacts on soil nutrients availability, and the use of numerical models to estimate soil physical properties is necessary to avoid time consuming and labor intensive soil measurements.

However, all soil physical properties predicted by SOILWAT model were significantly different from the measured values at all of the examined agro- ecological zones, which suggests that current version of the SOILWAT model cannot be applied to represent soil water characteristics. Extensive improvements are needed for the SOILWAT model before it can be used for irrigation planning. I suggest rejecting this manuscript at this time.

**Response:** The results did not show all soil physical properties predicted by SOILWAT model to be significantly different from measured values. In Lines 19-20, it was reported that the SOILWAT model using texture and salinity data recorded similar textural classes for all the agro-ecological zones studied. Also, in Lines 326-328, the results showed that the use of the model with texture and salinity data alone strongly predicted witling point for two agro-ecological zones. As such the study do not suggest that the current version of the SOILWAT model cannot be applied to represent soil water characteristics. However, Lines 328-329, has been properly modified to conclude that texture and salinity alone are not sufficient to predict soil water characteristics. Lines 329-330 has been adjusted to advise that variables such as organic matter, bulk density and gravel be included for accurate prediction of soil water characteristics.

It was stated in Line 15-16 that the model was assessed for its efficiency in predicting soil moisture characteristics using only soil texture and salinity data, as opposed to the use of soil texture and organic matter data which was validated to be sufficient for accurate prediction for soil moisture characteristics in the advent of limited data conditions by Saxton and Rawls (2006) (Lines 75-77). Although, this was not clearly stated in the text, it has now been articulated in the objective (Page 3, Line 84) and materials and methods (Page 6, Line 185-188) to clearly reflect the focus of the study.

**Specific comments**

**Query:**
Page 2, ln 37-39: Is there any reference that supports this statement? If so, please cite here.

**Response:**
The references to support the statements have been included in Page 2, Lines 36 and 38.

**Query:**
Page 3, ln 96: What type of grassy vegetation is predominant here? C3 or C4 photosynthesis pathway?

**Response:**
The grassy vegetation are predominantly those that assimilate carbon dioxide by the C4 photosynthetic pathway (Page 3, Line 97).

**Query:**
Page 4, ln 112: Please indicate the location of this agro-ecological zone as the format used in "Derived Savannah".

**Response:**
The location (Kogi State) of the agro-ecological zone has been indicated (Page 4, Line 113).

**Query:**
Page 4, ln 117: Please indicate the location of this agro-ecological zone as the format used in "Derived Savannah".

**Response:**
The location (Edo State) of the agro-ecological zone has been indicated (Page 4, Line 118).

**Query:**
Page 6, ln 181: Which type of the T-test is used in the analysis? One-tailed or two-tailed?

**Response:**
The type of T-test (two-tailed) has been included in Page 6, Line 184.
**Query:**
Page 7, ln 221-223: Please rephrase the sentence. The R-squared value (0.44) could be acceptable here, but it doesn't mean that SOILWAT model can be used to predict soil available water.

**Response:**
The sentence has been rephrased in line 226-228 as suggested by the reviewer.

**Query:**
Page 7, ln 225-228: If model performance relies highly on soil organic matter, why didn't the authors include this information in the SOILWAT model?

**Response:**
It was stated in Line 15-16 that the model was assessed for its efficiency in predicting soil moisture characteristics using only soil texture and salinity data, as opposed to the use of soil texture and organic matter data which was validated to be sufficient for accurate prediction for soil moisture characteristics in the advent of limited data conditions by Saxton and Rawls (2006) (Lines 75-77). Although, this was not clearly stated in the text, it has now been articulated in the objective (Page 3, Line 84) and materials and methods (Page 6, Line 185-188) to clearly reflect the focus of the study.

**Query:**
Page 8, ln 239-245: Why didn't include silt adjustments in SOILWAT model to improve its performance for bulk density?

**Response:**
The authors did not include appropriate local adjustments for silt because there is no provision for silt input and adjustments in the version of the SOILWAT model used.

**Query:**
Page 8, ln 255-260: Why didn't include appropriate local adjustments for soil organic matter to improve the model results?

**Response:**
The authors did not include appropriate local adjustments for soil organic matter to improve the model results because we were only considering texture and salinity variable adjustments in this assessment of the SOILWAT model.

**Query:**
Page 9, ln 272-278: I agree with the authors that soil density can largely affect soil hydraulic conductivity simulations, but can it cause an order of magnitude difference between the measured and predicted soil hydraulic conductivity? Was there anything wrong in the SOILWAT model configurations?

**Response:**
There was nothing wrong in the SOILWAT model configuration. I think that the exclusion of organic matter adjustments in this prediction study could have been responsible for this disparity as organic matter is a fundamental parameter that strongly influences soil hydraulic conductivity.

**Query:**
Page 10, ln 329-333: The authors claimed that additional variables can help improve the simulation results from SOILWAT model; however, no results were shown to support this statement. More efforts are needed before using the SOILWAT model to predict soil moisture characteristics for irrigation planning and scheduling.

**Response:**
The statement has been re-casted to make reference to earlier studies as intended by the authors in Page 11, Line 333,334 and 336.

**Query:**
Page 15, Table 1: Please correct the format for "moisture-conductivity" and "gravel effects" (center, bold, and underline).

**Response:**
The format for "moisture-conductivity" and "gravel effects" has been corrected (centralized, bold, and underlined) as suggested by the reviewer.

**Query:**
Page 21, Fig. 1: Please provide a better resolution figure to replace this one. It's hard to tell the characters in each shaded box.

**Response:**
Efforts to get a better resolution to replace this version of Figure 1 have been proved abortive.

NB: Due to the revision carried out on the manuscript, additional references have been added to the list of references.

**Appreciation**

I must sincerely appreciate your criticism, contributions and suggestions, which have tremendously improved this manuscript.

Thank you very much.

**OrevaOghene Aliku**
**Department of Agronomy,**
**University of Ibadan, Nigeria.**

---

## Author Comment (AC3) · 11 Oct 2016

1 Revision Note Journal Name: Geoscientific Model Development Manuscript Number: gmd-2016-165 Manuscript Title: Establishing relationship between measured and predicted soil water characteristics using SOILWAT model in three agro-ecological zones of Nigeria Dear Editor, Thank you very much for providing me with the opportunity to revise the manuscript. I have revised and improved all sections following your suggestions. NB: All revised areas have been highlighted in yellow colour in the annotated manuscript (manuscript tagged "Soil Water Characteristics Model Revised"). ANONY-

MOUS REFEREE #2 General comments: Soil water is a highly spatially and temporally variable. Determining soil water is quite difficult. Real, direct measurement on soil is valuable. This study use SOILWAT model compared to laboratory measurement of soil physical properties (soil available water, bulk density, field capacity, saturated hydraulic conductivity, soil moisture content, maximum water holding capacity, and wilting point) at three agro-ecological sites of Nigeria. However, the results showed that SOILWAT cannot act as a good model to present the real status in the case. The authors should pass their own special opinion to modify it. Otherwise, this work remained uncompleted. For example, in page 7, line 227-228, as Saxton and Rawls (2006) indicated the organic matter content is important, and then the author should make a contribution to resolve it. Investigate the impact of vegetation on soil water is also needed. Besides, the purpose of this study also wants to manage irrigation planning and scheduling, but only the comparison between model and laboratory measurement has been shown. Not have detailed discussion in the management. The study should take responsibility for the research community, how the works can make a contribution to the related topic should be included. Response: The authors suggest that silt adjustment be included to the SOILWAT model in Page 1, Line 25, and Page 12, Line367, respectively. Also, the relation of this study to irrigation has been improved upon as seen in Page 13, line 399 – 404. Some questions and suggestions: Query: The review in performance of SOILWAT is lacked. How do authors think SOILWAT is a good tool to do such research in this region? Authors should use the physics formula to explain the uncertainties of predicted parameters. The explanations are too weak. 2 Response: Review performance of SOILWAT and the build-up to the use of the model has been improved upon as can be seen in Page 3, Line 78-96. The explanations have been improved upon and the formulae have been included in some of the explanation (Page 9, Line 268 – 272; Page 10, Line 303 – 305; Page 12, Line 375 – 379) to lay emphasize on the importance of some of the parameters and the uncertainties of predicted parameters. Query: Page 4, line 87: Why do authors choose these sites? Response: The reasons for the choice of these sites are stated in Page 3, Line 88-99. Query: Page 4, line 112:

Lack of information as what in 'Derived Savannah' Response: The information for this Location has been improved upon as presented in Page 5, Line 133-139 Query: Page 4, line 117: Lack of information as what in 'Derived Savannah' It may need a real map to show the location of these three sites. Response: The information for this Location has been improved upon as presented in Page 5, Line 145-150. The Locations of these sites have been included as appeared in Figures 1, 2 and 3, respectively. Query: Page 7, line 222-223: The R-squared value (0.44) could not indicate that SOILWAT model can be used to predict soil available water. Response: The sentence has been rephrased in line 226-228 as suggested by the reviewer. Query: Page 9, line 279: The results showed SOILWAT overestimated at Derived savannah and rainforest but under estimated in Savannah in soil moisture content, how do authors comment it? Response: This sentence has been rephrased as suggested. Query: Page 21, Figure. 1: Please provide the description on it and also a better resolution one to replace it. 3 Response: Efforts to get a better resolution to replace this version of Figure 1 have been proved abortive. NB: Due to the revision carried out on the manuscript, additional references has been added to the list of references. Appreciation I must sincerely appreciate your criticism, contributions and suggestions, which have tremendously improved this manuscript. Thank you very much. OrevaOghene Aliku Department of Agronomy, University of Ibadan, Nigeria.